# Effects of Lipase and Xylanase Pretreatment on the Structure and Pulping Properties of Wheat Straw

**DOI:** 10.3390/polym14235129

**Published:** 2022-11-25

**Authors:** Qianqian Jia, Jiachuan Chen, Guihua Yang, Kefeng Liu, Yueying Wang, Kai Zhang

**Affiliations:** State Key Laboratory of Biobased Materials and Green Papermaking, Qilu University of Technology, Shandong Academy of Sciences, Jinan 250353, China

**Keywords:** wheat straw, bio-enzyme pretreatment, pulp properties, alkali-oxygen pulping, structure

## Abstract

Based on the reduction of environmental pollution, a biological enzyme assisted alkali-oxygen pulping method was explored to improve the delignification efficiency and fiber accessibility of wheat straw and improve the properties of wheat straw pulp. In this paper, lipase and xylanase were used to pretreat wheat straw and the effects of different enzyme types and enzyme dosage on the microstructure and pulp properties of wheat straw were investigated and experimented. The results showed that the lipase can remove fat and wax on the surface of wheat straw, while xylanase degraded the hemicellulose components, such as xylan, of wheat straw fiber, destroyed the structure of the lignin-carbohydrate complex, increasing lignin removal as a result and enhancing the impregnating, diffusion and penetration of alkali. Compared with wheat straw without enzyme pretreatment, the skeleton of wheat straw pretreated by enzyme became looser, the internal cavity appeared and the wall cavity became thin and transparent. The fines decreased obviously and the length of fibers increased. After combined pretreatment with lipase (15 U·g^−1^) and xylanase (15 U·g^−1^), the pulping performance of wheat straw was improved and the tensile index (97.37 N·m·g^−1^), brightness (40.9% ISO) and yield (58.10%) of the pulp increased by 12.9%, 19.9% and 9.9%, respectively. It can be seen that enzyme pretreatment is a green and effective approach to improving the alkali-oxygen pulping performance of wheat straw.

## 1. Introduction

Wheat straw, as one of the non-wood feedstocks, has abundant quantity in biomass resources. However, a massive amount of wheat straw is burned and landfilled because of its long-term storage, low density, high ash content, etc. [1], which causes a waste of resource and serious environmental pollution [2]. Recently, the pulp and paper industry are facing dilemmas, including the demand for paper products and the lack of lignocellulose [3]. The lack of applications of wheat straw offsets the sustainability advantages as a supplementary material for paper making [4]. Thus, it is essential to develop a large-scale and low-cost pulping approach to maximize the value of non-wood lignocellulose. This would not only solve the shortage of feedstocks, but also reduce environmental pollution and carbon emissions [5,6].

Due to a lower reaction temperature and less energy consumption [7], the alkali-oxygen method is considered to be cleaner than the conventional pulping process [8,9]. Under an alkali and oxygen pressure environment, non-phenol structures in lignin are easily depolymerized and broken into low molecular weight products [10], which leads to the fibers swelling with more charge and superior ductility. Additionally, silica in black liquor can be deposited on the fiber surface, improving the brightness and strength of pulp, which solves the problem of silicon interference in the alkali recovery process [11,12,13]. 

Nevertheless, many natural factors hinder effective delignification, such as strong intra- and inter-hydrogen interactions, tough wax surface, lignification degree, heterogeneity and complexity of cell wall composition structure, and complex crosslinking, which are defined as “biomass recalcitrance” [14,15]. Hence, it is necessary to increase the accessibility of wheat straw via pretreatment and maintain low usage of cooking reagent. At present, various pretreatment methods have been developed to promote lignocellulosic potential application, including steam explosion, extrusion, oxidation and enzyme pretreatment, etc. [16,17]. Among these, enzyme pretreatment featured high efficiency, freedom from contamination and low power consumption, and is considered a promising method [18,19]. Recently, different types of enzyme, including lipase and xylanase, were used to eliminate the specific substrate to optimize the pulping process. Lipase can hydrolyze triglycerides to partial triglycerides and fatty acids, loosening the fiber wall structure by depolymerization [20], which can increase the accessibility of subsequent reactants, facilitate delignification and improve the internal binding strength of fibers [21]. Xylanase is a glycosidic hydrolase with complex enzymatic systems, cutting off the linkage between lignin and carbohydrate by hydrolyzation, which improves the permeability of pulping chemicals, reduces the chemical load and increases the strength properties of pulp [22,23,24]. In addition, it is also free from water pollution, especially AOX formation [25]. A similar result was obtained from the synergistic treatment of enzymes. In the lipase-xylanase treatment process, xylanase hydrolyzes the glycosidic linkage to achieve delignification, while lipase removes fat and wax on the surface of wheat straw and increases the action area of xylanase. After synergistic pretreatment, the tensile strength increased by 20.96% and Kappa number decreased by 4.54% [26].

Based on the above consideration, herein a biological enzyme assisted alkali-oxygen pulping method was explored to improve the delignification efficiency and fiber accessibility of wheat straw and improve the properties of wheat straw pulp. The morphology, ultrastructure of wheat straw and pulp fibers were measured by scanning electron microscopy (SEM), micro-CT and other high-resolution imaging, along with Fourier transform infrared (FTIR), X-ray diffraction (XRD), X-ray photoelectron spectroscopy (XPS) and other spectral techniques. Additionally, the detailed structure changes of wheat straw and pulp fibers during the cooking process were investigated, which can provide enlightenment regarding the use of enzyme pretreatment, and thus improve pulp strength and reduce energy consumption and chemical reagents. The effects of enzyme types and enzyme dosage on the strength properties of pulp were also observed. In short, it is expected that an enzyme-assisted alkali-oxygen pulping process could effectively improve delignification and accessibility, which will enhance pulp properties and reduce environmental pollution.

## 2. Experimental Section

### 2.1. Materials

Wheat straw was provided by a paper mill in Shandong province. It was cut into 3~5 cm length and set aside after being cleaned, squeezed, twisted and air-dried. Part of the air-dried wheat straw was ground and a 40–60 mesh fraction was selected for chemical analysis. The chemical components of wheat straw were cellulose (36.36%), holocellulose (67.88%), lignin (19.20%), 1% NaOH extractives (43.34%) and ash (8.60%). The lipase and xylanase were obtained from Shandong Lonct Enzymes Co., Ltd. (Linyi, China) and sodium carbonate was purchased from Sinopharm Chemical Reagent Co., Ltd. (Shanghai, China).

### 2.2. Analysis of Enzyme

The activity of lipase was analyzed with olive oil emulsion [27] and 50 mL of olive oil was mixed with 50 mL of gum arabic solution (7%, *w/v*). The mixture contained 5 mL emulsion, 2 mL sodium phosphate buffer (pH 7.0) and lipase (1 mL), reacted at 37 °C for 5 min, adding 10 mL acetone-ethanol solution (1:1) to stop the reaction. The resulting fatty acids were titrated with 25 mM potassium hydroxide solution, with phenolphthalein as the indicator. The activity of lipase was determined to be 50,000 U·mL^−1^.

Xylanase activity was analyzed by incubating 1 mL of xylan and 0.5 mL of xylanase in an acetate buffer (pH 4.8) at 50 °C for 30 min. The activity of xylanase was indirectly expressed by the content of reducing sugar produced, and this reducing sugar was analyzed by DNS (3,5-dinitrosalicylic acid) reagent [28]. The xylanase activity was determined to be 10,000 U·mL^−1^. 

### 2.3. Bio-Enzyme Pretreatment of Wheat Straw

The wheat straw was pretreated with 10, 15 and 20 U·g^−1^ lipase at pH = 7 and 45 °C for 4 h, and rubbed every 30 min to make even contact with the enzyme solution. The wheat straw pretreated with lipase was recorded as WS_L10_, WS_L15_, and WS_L20_. Similarly, the wheat straw was pretreated with 10, 15, and 20 U·g^−1^ xylanase at pH = 5 and 55 °C for 4 h [29,30] and recorded as WS_X10_, WS_X15_, and WS_X20_. Finally, under the optimal enzyme dosage, the wheat straw was pretreated with compound enzyme (lipase and xylanase) in a definite sequence, recorded as WS_LX15_. The control group was WS_0_, which was not pretreated with any bio-enzyme.

### 2.4. Alkali-Oxygen Pulping Process

After pretreated with bio-enzymes, the wheat straw was mixed with 18% Na_2_CO_3_ solution and transferred to a reactor, then repeatedly inflated and exhausted until all air was discharged from the reactor, maintaining the initial oxygen pressure of 1.4 MPa. The reactor was fixed to the KLJX-2 homogeneous reactor (Shanghai Kelai Instrument and Equipment Co., Ltd., Shanghai, China), turned at the speed of 15 rpm for 3 min to make full contact with alkali and oxygen molecules, and kept at 135 °C for 2 h. The pulp obtained after pretreatment of wheat straw with different dosage of lipase was recorded as P_L10_, P_L15_, and P_L20_. Similarly, the pulp obtained after pretreatment of wheat straw with different concentrations of xylanase was recorded as P_X10_, P_X15_, and P_X20_. Compound enzyme and control pulp were denoted as P_LX15_ and P_0_. Meanwhile, the paper was made after beating (SK 90S/4 TF, NORD, Bargteheide, Germany).

### 2.5. Properties of Wheat Straw

The three-dimensional structure of wheat straw was observed by micro-CT (SkyScan2211, Bruker, Saarbrucken, Germany). Before observation, the wheat straw was cut into 10 mm × 1 mm × 1 mm (length × width × thickness) and fixed on the sample rod with plasticine. The test conditions of micro-CT were power supply voltage of 50 kV, source current of 350 μA, original focusing mode of microfocusing, exposure time of 500 ms and image pixel size of 300 nm. For the X-ray diffractometer (XRD, D8-ADVANCE, Bruker) test, the wheat straw powder was evenly spread on the amorphous glass sample table. The test conditions were as follows: diffraction angle 10~80°, scanning speed 20°/min, working voltage 40 kV.

### 2.6. Properties of Pulp

The pulp samples before and after pretreatment with bio-enzymes were used for the handsheet formation to test the strength properties. The pulp was used for 70 g/m^2^ hand sheets preparation and the kappa number of pulp was measured by the referenced standards methods (TAPPI T236 om-99, 2004) [31]. The hand sheets were used for testing tensile index (TAPPI T494 om-01, 2001) using XLW-B Tensile Strength Tester from Labthink Jinan (Jinan, China) [32], the tear index (TAPPI T414 om-04, 2004) and brightness (TAPPI T217 wd-77, 2004) using SLY-1000 Tearing Tester and YQ-Z-48B Brightness Tester from Hangzhou Qingtong & Boke Automation Technology Co., Ltd. (Hangzhou, China) [33,34]. 

### 2.7. Characterization of Fiber

The surface properties of pulp were analyzed by Scanning Electron Microscopy (SEM), Energy Dispersive Spectroscopy (EDS) and X-ray Photoelectron Spectroscopy (XPS). The pulp powder was pasted on the conductive adhesive and fixed to the sample stage. After spraying gold, SEM (Hitachi Regulus 8220, Tokyo, Japan) was used to observe the morphology of the pulp. The Hitachi Regulus 8220 with XFlash^®^ 6T-60 detector was used to observe the element distribution on the surface of pulp. For XPS (ESCALABXi+, Thermo Fisher Scientific, Waltham, MA, USA) analysis, the pulp powder was glued to the matte surface of aluminum foil paper with double-sided adhesive and the powder was pressed under 5–8 MPa for about 10 s. The aluminum foil paper with the sample was cut off and attached to the sample table for testing, and the XPS peak software was used to fit the peaks and process the data.

The length, width, kink index and fines of the fiber were measured by L&W fiber tester plus (FQA, 912+, Stockholm, Sweden) after the pulp was disentangled. The calculation method for curl index is shown in Formula (1) [35].


Curl index (%) = (1/shape factor) − 1(1)


The dried pulp was ground to powder and mixed evenly with KBr. After grinding and pressing, the pulp was tested by FTIR (ALPHA, Bruker). The test conditions were as follows: scanning speed 16 times/second, scanning range of 4000~500 cm^−1^. For X-ray diffractometer (XRD, D8-ADVANCE, Bruker) test, the pulp powder was evenly spread on the amorphous glass sample table. The test conditions were as follows: diffraction angle 10~80°, scanning speed 20°/min, working voltage 40 kV. The calculation method of crystallinity index was shown in Formula (2). 


CrI = (I_002_ − I_am_/I_002_) × 100%(2)


## 3. Results and Discussion

The lipids, waxy layers, lignin and other components of wheat straw provide a dense structure for the cell wall, which limits the effective separation of fiber components in the pulping process. In traditional pulping methods, the non-phenolic structure of lignin is difficult to degrade under alkaline conditions alone. However, in the process of alkali-oxygen pulping, the synergistic effect of reactive oxygen molecule and alkali is conducive to the degradation of benzoic structural intermediates after cleavage of C_α_-C_β_ bond in the side chain of lignin. At the same time, after the aryl-ether bond is hydrolyzed, the phenol structural intermediate is degraded, which promotes the degradation of the non-phenolic structure into a low molecular mass product, thereby promoting the removal efficiency of lignin [10]. Therefore, as shown in Figure 1, lipase and xylanase were used for pretreatment of wheat straw to destroy its dense structure, increasing the accessibility of oxygen and alkali to the substrate during alkali-oxygen pulping. The action mechanism of lipase and xylanase was mainly investigated through the physical strength, surface morphology, FQA, FTIR, XRD and XPS of pulp fiber, and the XRD and three-dimensional structure of wheat straw.

### 3.1. Pulp Properties

#### 3.1.1. Effect of Bio-Enzyme Dosage on the Properties of Pulp

Figure 2 shows the effect of different enzyme dosage on the properties of pulp. As shown in Figure 2a,b, P_L15_ exhibited the highest tensile index and tear index, which were 92.37 N·m·g^−1^ and 6.75 mN·m^2^·g^−1^, respectively. Similarly, the tensile index and tear index of P_X15_ reached the maximum, which were 91.11 N·m·g^−1^ and 7.15 mN·m^2^·g^−1,^ respectively. Significantly, the tensile index of P_LX15_ can reach 97.37 N·m·g^−1^, which was 11.12 N·m·g^−1^ higher than P_0_ (86.25 N·m·g^−1^), 5.00 and 6.26 N·m·g^−1^ higher than P_L15_ and P_X15_, respectively. As shown in Figure 2c, the brightness of the pulp was improved after bio-enzyme pretreatment. The P_X20_ displayed the highest brightness, which reached 44.8% ISO. The brightness of P_LX15_ reached 40.9% ISO, which was 6.8% ISO higher than that of P_0_ (34.1% ISO). From Figure 2d, we can see that the P_LX15_ showed the highest screened yield (58.10%), which was 5.24% higher than P_0_ (52.86%), and also higher than that of single bio-enzyme pretreatment. Compared with P_0_, the kappa number of P_LX15_ was slightly decreased (Figure 2e).

The increase of tensile index after lipase pretreatment may be due to the fact that most fat and wax on the surface of wheat straw were removed, reducing the penetration resistance of alkali and oxygen, accelerating the efficiency of the synergistic delinification, which is conducive to improving the moistening and swelling performance of fiber, and thereby improving the softness and plasticity of fiber, finally increasing the strength of pulp [36].

However, with the increase of enzyme dosage, the enzyme molecules permeate and act on the inside of the fiber cell wall, causing the tiny pores of the fiber cell wall to gradually expand, and these tiny pores are interconnected and transformed into larger pores. During the alkali-oxygen pulping, the microfiber fragments produced by biological polysaccharides and lignin polymers easily blocked the pores and hindered the subsequent action of lye [37], resulting in decreased physical strength. Xylan is a physical barrier, which can wrap cellulose microfibrils and limit cellulose fiber swelling. Xylanase acts on xylan, which promotes the easier breaking of the P and S_1_ layer of the wheat straw fiber cell wall, resulting in the separation of lignin from LCC [38]. This promoted the dissolution of lignin and the fiber hydration and swelling of the fibers, enhancing the binding effect between the fibers, thus improving the physical strength of the pulp. However, excessive xylanase will cut off the hydrogen bond between hemicellulose and cellulose, causing carbohydrates to dissolve, thereby reducing the binding strength between fibers [39]. The synergism between lipase and xylanase further improved the cellulose accessibility of the pretreated enzyme. The results showed that the combination of these strategies was suitable for improving the strength of the pulp while maintaining the properties of cellulose, without causing further cellulose degradation [40].

#### 3.1.2. Effect of Bio-Enzyme Pretreatment on Surface Morphology of Pulp Fibers

As shown in Figure 3, the pulp fibers exhibited a rough surface and intertwined with each other. In addition, there are many white particles attached to the surface and the inside of fibers. These white particles have a fairly uniform size distribution and can be nicely dispersed into the fiber layer. Through the EDS analysis, it can be seen that carbon and oxygen elements are uniformly distributed on the surface of the pulp, while silicon elements are concentrated. It can be confirmed that these white particles gathered silicon content, and may be silicate particles. Alkali-oxygen pulping can synchronously deposit silica on the surface of the cellulose to alleviate the problem of silicon interference in the black liquor. This may be due to the fact that, under the action of oxygen-containing free radicals, the dissolved lignin and degraded carbohydrates are further oxidized to organic acids and CO_2_, the pH of the system decreases and silicon is deposited on the fiber surface in the form of micro-nano silicate crystals, which effectively reduces the silicon content in the black liquid. In addition, silicate can be used as a carrier for oxygen molecules due to its large specific surface area, which improves the brightness of the pulp and effectively reduces the dosage of the bleaching agent in the bleaching section [41].

#### 3.1.3. Effect of Bio-Enzyme Pretreatment on Pulp Fiber Quality

Table 1 shows the morphological properties of fibers before and after bio-enzyme pretreatment. It can be observed that the fines content decreased after bio-enzyme pretreatment. This phenomenon can be attributed to the large specific surface area of fines, which will preferentially adsorb enzyme molecules [42]. Additionally, the content of relative long fibers increased, resulting in an increase in the average fiber length.

After lipase pretreatment, the fat and wax on the surface of the wheat straw were removed, making it easy for chemical reagents to penetrate into the inside of the pulp fibers, promoting the removal of lignin and improving the softness of the fibers. Therefore, the cutting of the fibers was reduced during beating and the fiberization, water absorption and swelling of the fibers were improved [43]. Additionally, the specific surface area of the fibers increased and more hydroxyl groups were exposed on the fiber surface, thus increasing the binding strength between the fibers. After xylanase treatment, the fiber length and aspect ratio increased, mainly because the part of the hemicellulose on the fiber surface and cell wall is easy to degrade, which is conducive to the absorption and expansion of the fiber, and is not easy to be cut off [44].

The fiber length of P_LX15_ was 0.806 mm, which was 0.095 mm longer than that of P_0_. Under the synergistic action of lipase and xylanase, the lipids of wheat straw can be removed, and the degradation of some hemicellulose components makes the fiber structure loose and porous, which is beneficial for the water absorption, swelling and chemical properties of the fiber [45]. At the same time, the larger fiber aspect ratio is conducive to increasing the amount of interweaving between fibers per unit area of the paper, making the fiber distribution more uniform, which can effectively fill the gaps between the fibers. However, the kink index and curl index of the pulp decreased after bio-enzyme pretreatment and fiber curls and kinks commonly show greater accessibility to bio-enzyme [46], which is beneficial to the water filtration performance and strength of the pulp. Notably, appropriate amount of kink can improve tear strength, but too much kink will reduce the tensile strength of the pulp [47].

#### 3.1.4. FTIR Analysis of the Bio-Enzyme Pretreated Pulp

FTIR spectroscopy was applied to characterize the functional groups change of cellulose and lignin in different pulps. As shown in Figure 4, the absorption peaks were classified according to [48]. The absorption structure of P_L15_, P_X15_ and P_LX15_ was similar to that of P_0_, but the intensity is different. An increase in transmission at 3420 cm^−1^ (-OH stretching vibration) was observed after bio-enzyme pretreatment, and the peak intensity of P_LX15_ was the highest. The increase in peak intensity after bio-enzyme pretreatment can be attributed to the increase of free hydroxyl groups in the pulp. The increase of free hydroxyl group can increase the number of hydrogen bonds within and between fibers and improve the pulp binding strength. In addition, the content of alcohol hydroxyl groups was increased, possibly due to the cleavage of adipo-ether bonds. Xylanase disrupts the LCC linkage [49], resulting in the increase of phenolic hydroxyl groups and alcohol hydroxyl groups at the C_β_ and C_α_ in the lignin structural unit. 

1433 cm^−1^ and 1335 cm^−1^ are characteristic peaks of lignin, corresponding to aromatic ring skeleton vibration. The bands at 898, 1050 and 1370 cm^−1^ correspond to glycosidic bond and C1 deformation vibration, C=O vibration and C-H bending vibration in carbohydrates in pulp. There was no significant change in the peak intensity, indicating that the cellulose structure would not be destroyed by bio-enzyme pretreatment.

#### 3.1.5. Crystallinity Analysis of the Bio-Enzyme Pretreated Pulp

X-ray diffraction (XRD) was used to evaluate the changes in the relative crystallinity index (CrI) of pulp by bio-enzyme pretreatment [50]. The diffraction patterns for all pulp consisted of well-defined peaks assigned to the (200) plane at 2θ = 22.3° and the (110) crystal planes at 2θ = 16°, and bio-enzyme pretreatment did not change the crystal shape of cellulose. Figure 5a show that the CrI of P_L15_, P_X15_ and P_LX15_ were 61.50%, 60.89% and 60.38%, respectively, which was slightly lower than that of P_0_ (62.36). This might be attributed to the fact that lipase pretreatment destroyed the fat and wax layer on the surface of wheat straw, reduced the resistance of alkali and oxygen penetration, accelerated the removal efficiency of lignin, and partially destroyed the cellulose crystallization region. After xylanase pretreatment, the hemicellulose content of the pulp decreased, which meant that the presence of amorphous components decreased [51,52].

#### 3.1.6. XPS Analysis of the Chemical Composition of the Bio-Enzyme Pretreated Pulp Fiber Surface

XPS determines the composition of chemical elements according to the change of chemical displacement of the electron binding energy on the fiber surface, which has been widely used to study the surface chemistry of lignocellulose [53]. As shown in Figure 6, C1 is 284.6 eV, corresponding to C-C/C-H, C2 is 286.1 eV, corresponding to C-O and C3 is 287.4 eV, corresponding to C=O/O-C-O [54]. C1 mainly reflects the non-carbohydrate content, such as lignin and extracts, whereas C2 and C3 mainly reflect the content of carbohydrates. The theoretical O/C values of cellulose and lignin are 0.83 and 0.33, respectively [55].

Compared with Figure 6a, after the bio-enzyme pretreatment, the intensity of C-C/C-H of the pulp decreased significantly, and the intensity of C-O and C=O/O-C-O increased significantly as seen in Figure 6b–d. As shown in Table 2, the values of C1/C2 of the pretreated pulp were significantly lower than that of the untreated pulp (1.03) and the values of O/C of the pretreated pulp were significantly higher than that of the untreated pulp (0.49). Xylanase acts on hemicellulose, opening its hydrogen bonding and releasing the chromogenic groups associated with lignin [56]. The natural recalcitrance of lignin was weakened and further restriction sites were exposed, which facilitate the subsequent penetration of alkali and oxygen [57]. The O/C of P_LX15_ was 0.61, which was slightly lower than P_L15_ and P_X15_. This might be due to the pretreatment of wheat straw by lipase, where a large amount of extract was removed in order to increase the accessibility of subsequent xylanases. Xylanase acts on the hydrogen bonds between hemicellulose and cellulose and the ether bonds between hemicellulose and lignin, which can promote the dissolution of lignin and the hydration, swelling and brooming of fibers. Moreover, the dissolved lignin was partially adsorbed and deposited on the surface of the fiber to prevent serious degradation of carbohydrates [58].

### 3.2. Effect of Bio-Enzyme Pretreatment on Wheat Straw Microstructure 

#### 3.2.1. Crystallinity of Wheat Straw

Figure 5b shows the peaks of crystalline (I_002_) and amorphous (I_am_) cellulose with different intensities [59]. The I_002_ intensities at 2θ = 22.3 represent crystalline cellulose and I_am_ intensities at 2θ = 16° represent amorphous cellulose. The crystallinity of WS_L15_, WS_X15_ and WS_XL15_ was 55.31%, 53.69% and 53.56%, which was higher than that of WS_0_ (50.68%), and the diffraction peak intensity at 22.3° increased. This was mainly due to the removal of cellulose disordered regions and amorphous hemicellulose during bio-enzymatic pretreatment. This plays a role in swelling and softening the amorphous region of fiber, so the response value and crystallinity of the crystalline region were increased [60].

#### 3.2.2. Three-Dimensional Tissue Structure of Wheat Straw

Micro-CT can be used to observe the three-dimensional internal structure of wheat straw. The internode of wheat straw is a hollow cylinder, which is composed essentially of the parenchymatous ground tissue, the vascular bundles, and the epidermis [61]. The outermost layer is the dense epidermis, which gives a certain mechanical strength. The denser internal layer is the vascular bundle tissue, which is composed of vascular bundle cells. Close to the medial vascular bundle cells are parenchymatous cells, which are larger in size and relatively minor in density [62]. 

As shown in Figure 7a,c, the fiber skeleton of wheat straw was complete, the outer side was relatively dense and the surface was covered by a smooth, dense and heavy film. This film is a stratum corneum composed of fatty and waxy layers, which hinders the infiltration of alkali. The diameter and volume of vascular bundle cells were smaller, the wall lumen was thicker and the inner wall was rough. The surface structure of the wheat straw is looser in Figure 7b,d. Under the action of lipase, a large amount of wax on the surface of the wheat straw was dissolved, the surface texture and pores gradually appeared and the vascular bundle cells on the inside and outside were partially degraded. The diameter of the vascular bundle became larger, the wall cavity became thinner, and the cell cavity of the parenchymatous became more transparent; these changes can provide favorable conditions for the penetration of chemicals. After lipase treatment, the composition of lipophilic substances in the waxy layer of wheat straw was effectively degraded and the surface hydrophilicity and wettability of wheat straw were improved, which is conducive to improving the pulping performance of wheat straw, meaning that wheat straw pulp can produce high-performance paper-based materials and improve the utilization value of wheat straw [63].

The pores of wheat straw in Figure 7e,g were more transparent and clearer than in Figure 7a,c. Xylanase can enter the interior through the breathing holes on the surface, degrading hemicellulose connected to vascular bundle tissue, parenchymatous tissue and fibrous tissue. The decomposition of tissue cells led to the exposure of voids, which was conducive to the expansion and softening of the internal fibers of wheat straw [64]. After xylanase treatment, the structure of wheat straw was loose and porous and part of the lignin was removed, which can be used to prepare ultra-strong and light-transmission film materials [65]. The fibers on the surface of the wheat straw in Figure 7f,h are distinctly visible, the skeleton looser than Figure 7a,c. Most of the vascular bundle cells on the inside and outside were degraded and the wall cavity became thinner and more transparent. These results indicated that the compound enzyme promoted the degradation of vascular bundle tissue fibers, which was beneficial to the adequate moisturize and swelling of fibers, thus reducing the fiber damage during pulping.

## 4. Conclusions

In this paper, the pretreatment of lipase removed the hydrophobic and lipophilic extractives and silica on the outer surface of wheat straw and then the pretreatment of xylanase further degraded xylan hemicellulose, which broke the structure of lignin-carbohydrate complex. Hereby the fiber skeleton of the enzymic pretreated wheat straw became looser and more porous and the wall cavity became thinner, which is beneficial to increasing the pore structure, improving the surface hydrophilicity and wettability of wheat straw and promoting the diffusion and impregnation of alkali and oxygen molecules during the cooking process. Enzymatic pretreatment changed the fiber morphology and microstructure of the pulp, and decreased the consumption of active alkali. The pulping properties of wheat straw and the strength properties of pulp have been improved and pulp can be used to produce high-performance paper-based materials, which makes the alkali-oxygen pulping of wheat straw industrially and economically feasible. The optimal enzymes dosage was 15 U·g^−1^ of lipase and 15 U·g^−1^ of xylanase in the enzymic pretreating process. Compared with the unpretreated pulp (86.25 N·m·g^−1^ of tensile index, 34.1% ISO of brightness and 52.86% of yield), the tensile index, brightness and yield of the pulp pretreated by compound enzyme can reach up to 97.37 N·m·g^−1^, 40.9% ISO and 58.10%, an increase by 12.9%, 19.9% and 9.9%, respectively, higher than the lipase pretreatment (92.37 N·m·g^−1^, 37.6% ISO and 54.54%) and the xylanase pretreatment (91.11 N·m·g^−1^, 39.0% ISO and 53.16%). The content of fines decreased obviously and the length of pulp fiber increased after bio-enzyme pretreatment, which is beneficial to the improvement of the drainability of pulp.

However, it should be noted that the immobilized technology of bio-enzymes is still immature and the pretreatment time is long, which limits continuous production. Moreover, the molecular structure and action mechanism of enzymes need to be further studied. Therefore, we want to improve the low utilization rate of enzymes and long processing time through microcapsule packaging, immobilization or modification of enzymes and optimization of reaction media. At the same time, bio-enzymes can be used as the model substrate to create a related enzymatic kinetic model and explore their mechanism of action in depth.

## Figures and Tables

**Figure 1 polymers-14-05129-f001:**
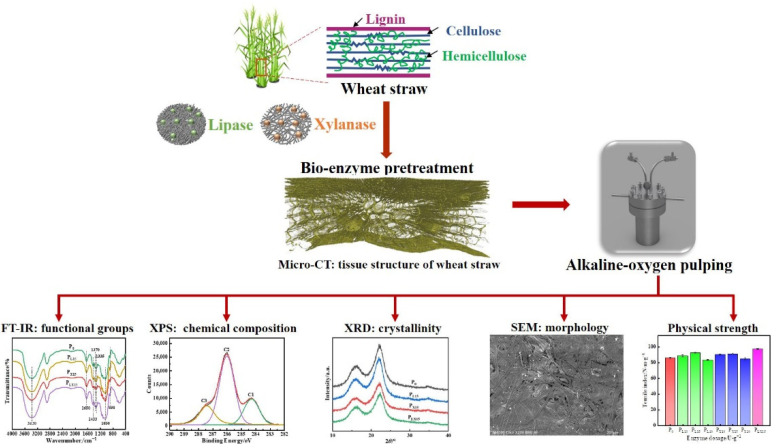
Schematic illustration of alkali-oxygen pulping of wheat straw pretreated with bio-enzyme.

**Figure 2 polymers-14-05129-f002:**
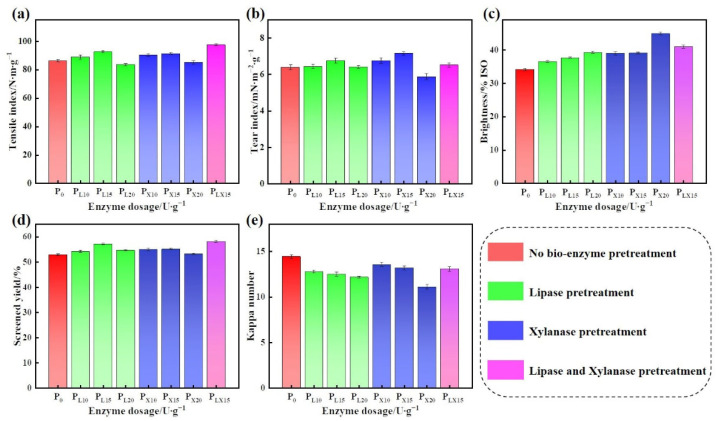
The effect of various bio-enzyme dosage on pulp properties: (**a**) Tensile index; (**b**) Tear index; (**c**) Brightness; (**d**) Screened yield and (**e**) Kappa number.

**Figure 3 polymers-14-05129-f003:**
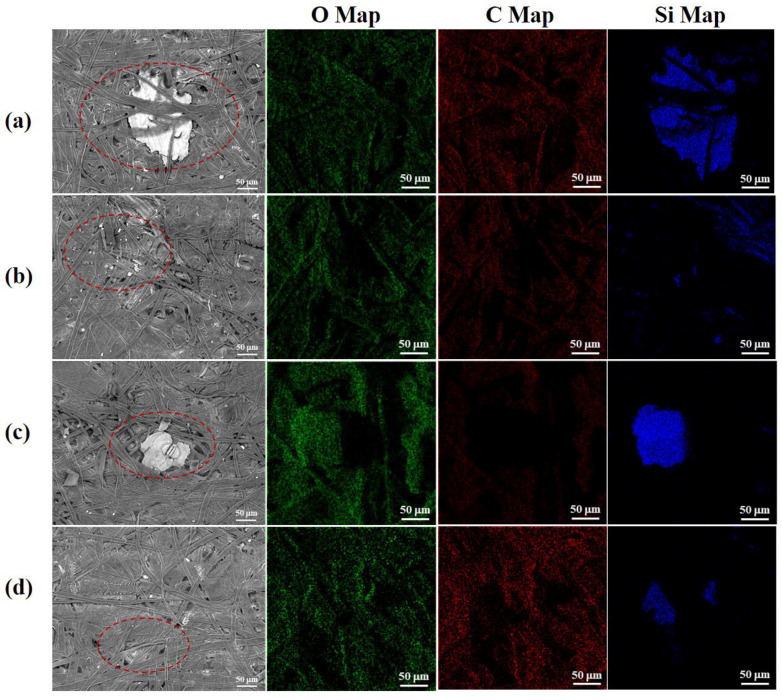
SEM images of pulp and distribution maps of elements on the outer surface of pulp: (**a**) no bio-enzyme pretreatment; (**b**) lipase pretreatment; (**c**) xylanase pretreatment and (**d**) lipase and xylanase compound enzyme treatment.

**Figure 4 polymers-14-05129-f004:**
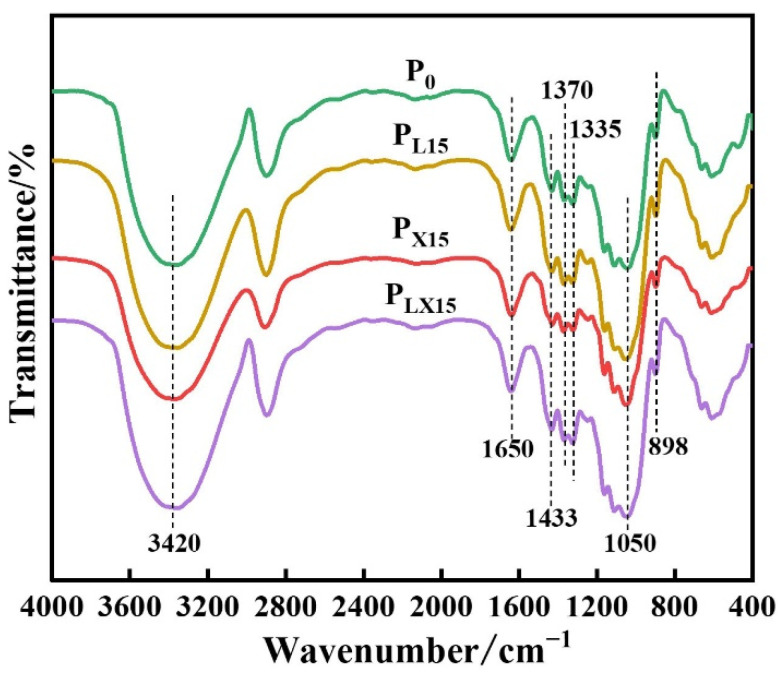
FTIR spectra of pulp before and after bio-enzyme pretreatment.

**Figure 5 polymers-14-05129-f005:**
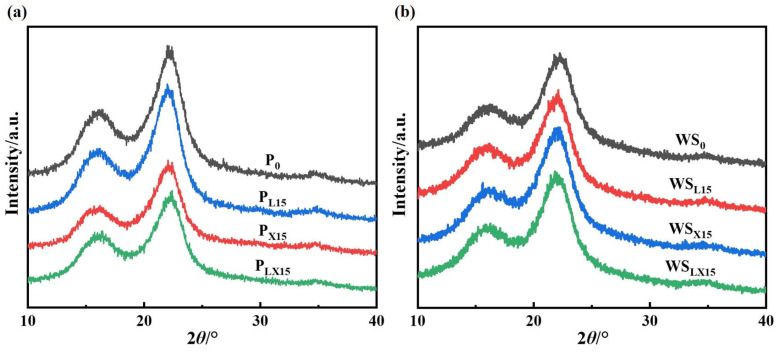
XRD patterns: (**a**) pulp and (**b**) wheat straw before and after pretreatment with bio-enzyme.

**Figure 6 polymers-14-05129-f006:**
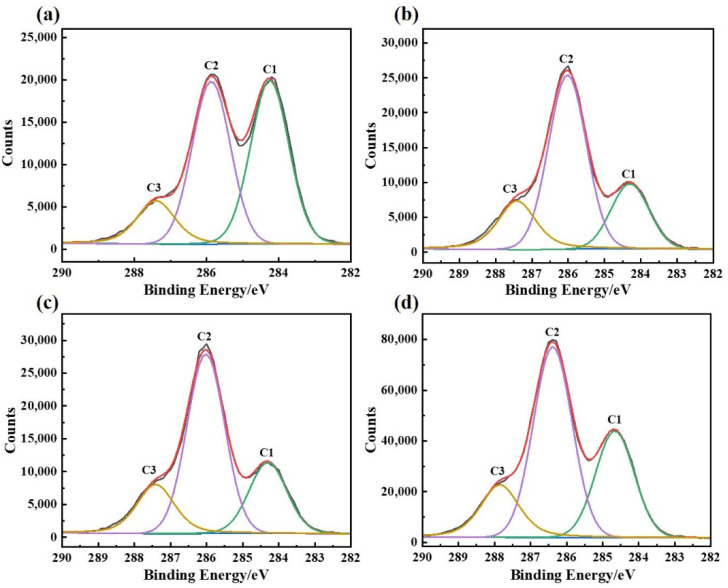
C1s spectra of pulp: (**a**) no bio-enzyme pretreatment; (**b**) lipase pretreatment; (**c**) xylanase pretreatment; (**d**) lipase and xylanase compound enzyme treatment.

**Figure 7 polymers-14-05129-f007:**
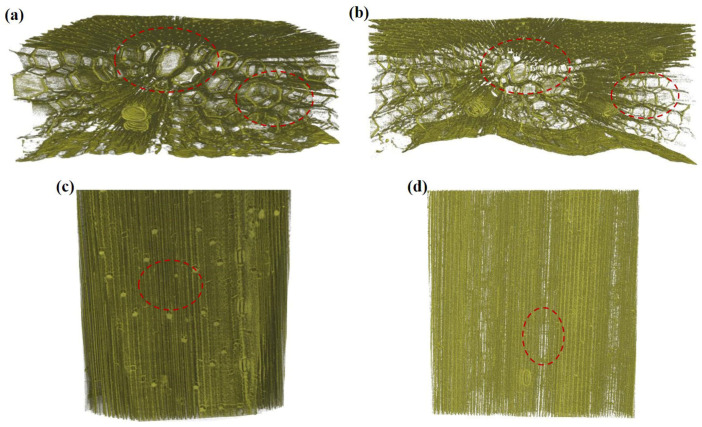
Micro-CT images of wheat straw: (**a**,**c**) no bio-enzyme pretreatment; (**b**,**d**) lipase pretreatment; (**e**,**g**) xylanase pretreatment; (**f**,**h**) lipase and xylanase compound enzyme treatment.

**Table 1 polymers-14-05129-t001:** Fiber characteristics of pulp before and after bio-enzyme pretreatment.

Pulp	Length /mm	Width /μm	Aspect Ratio	Fines /%	Kink Index	Shape Factor /%	Curl Index
P_0_	0.711	20.6	34.5	50.7	1.695	88.6	0.129
P_L15_	0.751	19.6	38.3	50.1	1.347	90.0	0.111
P_X15_	0.746	19.8	37.7	49.1	1.252	90.2	0.109
P_LX15_	0.806	19.7	40.9	49.3	1.060	90.8	0.101

**Table 2 polymers-14-05129-t002:** Pulp surface analysis.

Pulp	O 1s /%	C 1s /%	C1 /%	C2 /%	C3 /%	C1/C2	O/C
P_0_	32.38	66.24	43.97	42.51	13.52	1.03	0.49
P_L15_	38.66	60.48	21.57	58.36	20.07	0.37	0.64
P_X15_	39.36	59.86	23.70	57.75	18.55	0.41	0.66
P_LX15_	37.56	61.70	29.71	52.69	17.60	0.56	0.61

## Data Availability

Not applicable.

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
