# Peer review of "Effects of Lipase and Xylanase Pretreatment on the Structure and Pulping Properties of Wheat Straw"

_polymers, 2022, doi:10.3390/polym14235129_

Round 1

Reviewer 1 Report

Review report:

Compared with wheat straw without enzyme pretreatment, the skeleton of wheat straw pretreated by enzyme became looser, the internal cavity appeared, and the wall cavity became thin and transparent. The fines decreased obviously and the length of fibers increased. After combined pretreatment with lipase and xylanase.

Following are the comments after

Comment 1:

There is no specific description of its application

Comment 2:

 Its known that lipase and xylanase pretreatment, effects on the structure and pulping properties of wheat straw

DOI:10.1016/j.indcrop.2009.05.009

DOI:10.1016/S0141-0229(02)00050-9

Except combining them what is the main significance? The mechanism of combination that led to improvement is missing. Can be included in the conclusion or results section.

Comment 3:

The optimal enenzymeosage was 15 U.g-1 of lipase and 15  U.g-1 of xylanase in the enzymic pretreating process of wheat straw.

How did you come to this conclusion? What is the procedure for optimizing the doses? Does it depend on the doses you have considered or generated statistically?

Comment 4:

There is no specific activity reported about the enzymes during the treatment process.

Reviewer 2 Report

- In section 2.4 and 2.5 please add the some of the important properties of wheat straw and pulp respectively. 

- Add more explanations for Figures 1 & 3. 

- Write more about the Pulp surface analysis.

-  Include the limitations of the present work. 

- Include the future scope of the work. 

- Please illustrate the experimental procedures in detail. 

Round 2

Reviewer 1 Report

After revision the manuscript looks good now and, the manuscript is accepted.